# Neuron-Level Linguistic Selectivity in LLMs via a Classifier-Free Framework

**Linyang He**
Zuckerman Mind Brain Behavior Institute
Columbia University
`linyang.he@columbia.edu`

**Nima Mesgarani**
Zuckerman Mind Brain Behavior Institute
Columbia University
`nima@ee.columbia.edu`

**Editors:** Marco Fumero, Clementine Domine, Zorah Lähner, Irene Cannistraci, Bo Zhao, Alex Williams

## Abstract

Understanding how Large Language Models (LLMs) encode linguistic structures remains a fundamental challenge in interpretability research. While diagnostic classifiers (or "probes") are the standard tool for this task, they face significant methodological criticism: training auxiliary classifiers introduces capacity confounds and calibration issues, often making it difficult to distinguish the model's intrinsic representations from the probe's ability to learn the task. To address these limitations, we introduce a probe-free framework for localizing linguistic selectivity at the individual neuron level. Leveraging the controlled contrasts of linguistic minimal pairs, we propose Minimal-Pair Neuron Separability (MPNS), a metric that directly quantifies how reliably single neurons differentiate grammatical from ungrammatical constructions without parameter updates. By applying this framework to the Qwen3 model, we uncover a distinct functional hierarchy: syntactic and morphological processing is concentrated in early-to-mid layers, whereas semantic-syntactic interfaces and conceptual reasoning emerge in deeper layers . Furthermore, hierarchical clustering of sensitive neurons reveals a modular internal organization, identifying both domain-specific "specialists" and domain-general "integrators". Our approach yields fine-grained, interpretable maps of linguistic competence, offering a rigorous alternative to probing for mechanistic analysis.

## 1 Introduction

Where in a language model are specific syntactic distinctions represented, and can we organize the responsible *neurons* into coherent functional groups? Prior work has evaluated syntactic competence via targeted test suites and minimal pairs [15, 24, 4, 6, 7], and has analyzed the evolution of linguistic information across layers [11, 22]. However, standard *probing* techniques require training auxiliary classifiers, introducing capacity and calibration confounds [10, 20].

We propose a simple, *probe-free* neuron-level approach that leverages the controlled contrasts in BLiMP [24] and COMPS [18] to directly score each neuron's selectivity to a given linguistic phenomenon. For each minimal pair we extract last-token activations per neuron, assemble positive and negative activation vectors, and compute a correlation-derived separability index. A higher value of this index indicates that the neuron more clearly separates the positive from the negative members, suggesting that it encodes the syntactic phenomenon distinguished by the minimal pair. Conversely, a low index implies that the neuron's activations do not reflect this contrast, and thus the neuron is irrelevant or insensitive to the corresponding phenomenon.

Proceedings of the III edition of the Workshop on Unifying Representations in Neural Models (UniReps 2025).

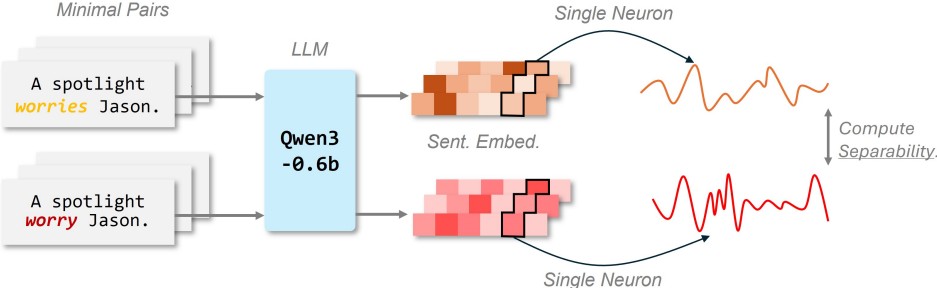

Figure 1: Illustration of the minimal-pair neuron separability pipeline. Minimal pairs from BLiMP/COMPS are fed into a transformer language model (here, Qwen3-0.6B). For each neuron, we extract last-token activations for the grammatical (positive) and ungrammatical (negative) members, assemble paired activation vectors, and compute a correlation-based separability score. High separability indicates that the neuron reliably differentiates acceptable from unacceptable sentences, providing a fine-grained map of linguistic selectivity at the neuron level.

Aggregating per layer yields a concise layer-wise selectivity curve; neurons at the peak layers are further organized via hierarchical clustering [23, 19, 17] to expose structure among linguistic phenomena. The result is an interpretable map of *where* and *which* neurons carry specific syntactic distinctions, consistent with emerging mechanistic accounts of transformer internals [3, 5, 16]. Our contributions are: **1) Minimal-pair neuron separability (MPNS)**: a correlation-based, probe-free measure of neuron-level discrimination for grammatical vs. ungrammatical contrasts. **2) Layer-wise selectivity and neuron-level category profiling**: curve and heatmap summarizing where syntactic distinctions concentrate in the stack across paradigms. **3) Neuron clustering of syntactic phenomena**: hierarchical clustering of peak-layer neurons reveals groups aligned with both "domain-specific" constraints for concept, semantics, syntax and "domain-general" functions.

## 2 Methods

Model and dataset details can be found in the appendix.

### 2.1 Setup and notation

Let $\mathcal{D}_p = \{(x_i^+, x_i^-)\}_{i=1}^{M_p}$ denote the minimal pairs for a BLiMP/COMPS paradigm $p$ (e.g., subject–verb agreement). For a transformer with $L$ layers and hidden width $d_\ell$ at layer $\ell$, let $h_\ell(x) \in \mathbb{R}^{d_\ell}$ be the last-token hidden state (for causal LMs). For neuron $j \in \{1, \ldots, d_\ell\}$ define scalar activation $a_{\ell j}(x) = h_\ell(x)_j$.

We z-score activations neuron-wise within each paradigm to remove scale:

$$\tilde{a}_{\ell j}(x) = \frac{a_{\ell j}(x) - \mu_{\ell j}}{\sigma_{\ell j}}, \quad \mu_{\ell j}, \sigma_{\ell j} \text{ are computed over } \{x_i^+, x_i^-\}_{i=1}^{M_p}.$$

### 2.2 Minimal-pair neuron separability

For neuron $(\ell, j)$ in paradigm $p$, form paired activation vectors

$$\mathbf{v}_{\ell j,p}^+ = [\tilde{a}_{\ell j}(x_1^+), \ldots, \tilde{a}_{\ell j}(x_{M_p}^+)], \quad \mathbf{v}_{\ell j,p}^- = [\tilde{a}_{\ell j}(x_1^-), \ldots, \tilde{a}_{\ell j}(x_{M_p}^-)].$$

Define the *Minimal-Pair Neuron Separability* score

$$S_{\ell j}(p) = 1 - \frac{\operatorname{corr}\left(\mathbf{v}_{\ell j,p}^+, \mathbf{v}_{\ell j,p}^-\right) - (-1)}{2} = \frac{1 - \operatorname{corr}\left(\mathbf{v}_{\ell j,p}^+, \mathbf{v}_{\ell j,p}^-\right)}{2}, \tag{1}$$

Intuitively, if a neuron responds differently to grammatical versus ungrammatical members across items, the correlation between their activations is low, yielding a separability score $S_{\ell j}(p)$ close to 1; conversely, if the activations are similar, the correlation is high and $S_{\ell j}(p)$ approaches 0. This paired sensitive index is scale-invariant (post z-scoring) and uses only controlled contrasts, avoiding classifier capacity issues.

## 2.3 Layer-wise selectivity

For paradigm $p$, the layer-wise selectivity is $\mathrm{LMS}_\ell(p) = \frac{1}{d_\ell}\sum_{j=1}^{d_\ell} S_{\ell j}(p)$. We average across paradigms within each BLiMP supercategory (e.g., *Agreement*, *Anaphor Agreement*, *Island Effects*) to obtain smoother curves. Because different syntactic and semantic phenomena can naturally induce distinct activation scales in the hidden states, a direct comparison across paradigms may otherwise be dominated by raw magnitude differences rather than genuine selectivity patterns. We therefore apply neuron-wise $z$-scoring within each paradigm, which removes scale disparities and ensures that the resulting curves reflect relative separability strength on a comparable basis across linguistic features.

## 2.4 Clustering neurons into linguistic groups

To compare neurons across paradigms within a common representational stage, we fix a single analysis layer $\bar\ell$ for all paradigms. Concretely, we set $\bar\ell = 15$, which is the mode of the peak layers obtained from the layer-wise mean separability curves; i.e., most paradigms exhibit maximal selectivity around layer 15. Fixing the layer allows us to analyze the *same* set of neurons across paradigms rather than mixing layers.

For each paradigm $p$, let $S_{\bar\ell j}(p)$ denote the separability score of neuron $j$ at layer $\bar\ell$. We select the top-$K$ sensitive neurons by separability at this fixed layer,

$$\mathcal{N}_p^{\mathrm{top}} = \left\{ (\bar\ell, j) \,:\, S_{\bar\ell j}(p) \text{ ranks among the top } K \text{ at layer } \bar\ell \right\},$$

with $K{=}5$ in our main analyses. We form the union of selected neurons across paradigms, $\mathcal{U} = \bigcup_p \mathcal{N}_p^{\mathrm{top}}$, which yields a set of unique neurons at layer $\bar\ell$. Given an ordered list of paradigms $\mathcal{P}$, we construct a phenomenon-by-neuron matrix $\mathbf{X} \in \mathbb{R}^{|\mathcal{P}| \times |\mathcal{U}|}$ with entries

$$X_{p,(\bar\ell,j)} = S_{\bar\ell j}(p).$$

We then cluster *neurons* (columns of $\mathbf{X}$) while keeping the paradigm order fixed on rows. Specifically, we compute pairwise *correlation distance*

$$D\big((\bar\ell, j), (\bar\ell, j')\big) = 1 - \mathrm{corr}\big(\mathbf{X}_{:,(\bar\ell,j)}, \mathbf{X}_{:,(\bar\ell,j')}\big),$$

and perform agglomerative hierarchical clustering with *average linkage* (UPGMA[21]). The dendrogram determines the column order. The reordered heatmap of $\mathbf{X}$ reveals coherent neuron groups at layer $\bar\ell$ with similar cross-paradigm selectivity profiles, aligning neurons with specific linguistic phenomena.

## 3  Results and Discussion

**Layer-wise selectivity.**   Figure 2-(a) shows the layer-wise mean separability (LMS) curves across linguistic forms. A clear peak emerges around layer15, where many grammatical categories, including morphology and core syntax, exhibit their strongest selectivity, consistent with the role of middle layers in encoding local grammaticality. Interestingly, we observe an additional peak around layer19, which is driven primarily by *Concept* contrasts. This suggests a functional separation: mid layers concentrate on structural well-formedness, while deeper layers shift toward conceptual and semantic–syntactic interface phenomena.

**Neuron Category Analysis**   Across layers, Figure 2-(b) distinct domains of linguistic selectivity emerge. Morphological and syntactic phenomena are most prominent in the lower and middle layers, consistent with their role in early structure-building and local grammatical constraints. In contrast, semantics-syntax interface phenomena (e.g., control/raising, quantifier scope, NPI licensing) and conceptual distinctions appear predominantly in later layers, reflecting higher-level interpretive operations. This developmental trajectory aligns with prior studies of representational progression in LMs [22, 11, 3].

Interestingly, neurons sensitive to semantics-syntax interfaces are concentrated in the later two-thirds of the model, but they do not saturate the entire layer. Instead, such neurons occupy localized subsets, suggesting that high-level interpretive phenomena are implemented by specialized subspaces rather than being distributed across all units. One plausible explanation is that interface distinctions require coordinated patterns across a subset of neurons, leaving other neurons available for orthogonal functions such as discourse modeling or world knowledge integration. This specialization might highlight an emergent modularity within deep layers of the model.

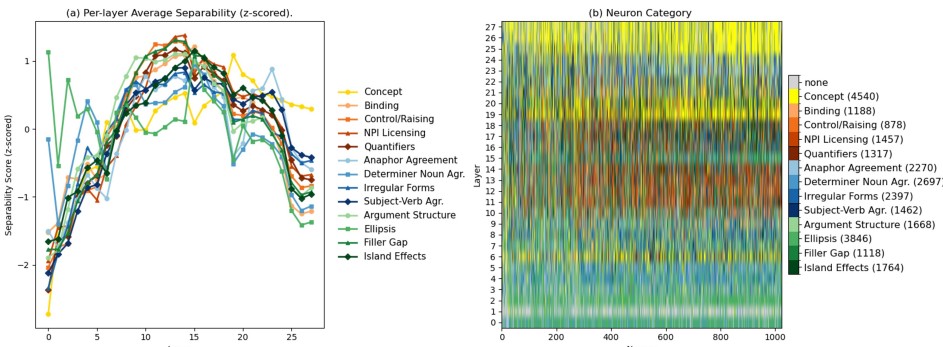

Figure 2: Colors indicate linguistic domains: *Concept*, *Semantics–Syntax Interface*, *Morphology*, and *Syntax*. (a) Layer-wise average separability (LMS) curves across linguistic forms. Each curve shows the mean separability of neurons within a form-specific paradigm, averaged across paradigms in the same BLiMP supercategory. Scores are $z$-scored within each paradigm to remove scale differences across phenomena, enabling direct comparison of relative selectivity patterns. (b) Neuron category assignment across layers. Each neuron is assigned to the linguistic paradigm with the highest $z$-scored separability (argmax), or labeled *none* if no form exceeds the threshold (z>0).

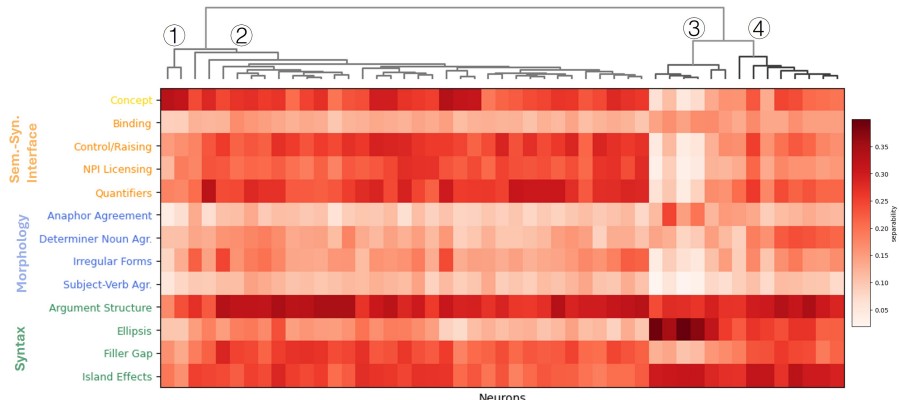

Figure 3: Hierarchical clustering of neurons at the fixed analysis layer ($\ell$=15). For each paradigm, the top-5 neurons by separability were selected and assembled into a phenomenon-by-neuron matrix. Columns (neurons) were clustered using correlation distance and average linkage, while rows (phenomena) were kept fixed and grouped by domain. The heatmap shows separability profiles with the dendrogram on top, revealing four major neuron clusters.

**Neuron clustering.**    Hierarchical clustering exposes four major groups of neurons with distinct selectivity patterns n Figure 3. Cluster 1 contains neurons that are strongly selective for Concept contrasts while remaining less sensitive to grammar, indicating highly domain-specific conceptual units. Cluster 2 consists of neurons responsive to both semantics and syntax, but with stronger weights toward semantic/Conceptual contrasts, suggesting mixed-domain but semantically biased encoding. Cluster 3 shows the opposite profile: these neurons are insensitive to semantics but highly responsive to syntactic and some morphological distinctions, reflecting grammar-specialized selectivity. Finally, Cluster 4 also integrates both domains but leans toward syntax, revealing generalist neurons with a structural bias.

Taken together, the clustering reveals a mixture of domain-specific neurons (e.g., conceptual specialists in Cluster1 and grammatical specialists in Cluster3) and domain-general neurons (Clusters2 and 4). This organization suggests that LLMs might not localize linguistic phenomena into completely disjoint sets of neurons; instead, they develop specialized subpopulations alongside integrative units. One possible explanation is that compositional language processing requires both dedicated circuits for sharp distinctions (e.g., anaphor agreement or conceptual categorization) and integrative circuits that bridge syntax and semantics for higher-level interpretation.

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

# A    Supplementary Content

## A.1    Models

We conduct our analysis on the recent **Qwen3-0.6B** model [25], a 0.6-billion-parameter causal transformer released by Alibaba Group. Qwen3 adopts a decoder-only architecture with rotary position embeddings, multi-head self-attention, and feed-forward layers following the standard transformer design. Despite its relatively modest size, Qwen3-0.6B achieves strong performance across a wide range of language modeling and reasoning benchmarks, making it a suitable testbed for fine-grained interpretability studies. Its compact scale allows us to efficiently extract and analyze neuron-level activations across all layers, while still reflecting representational trends found in larger-scale models.

## A.2    Dataset

We leverage two complementary minimal-pair resources. First, the **BLiMP** benchmark [24] provides 67 paradigms targeting core syntactic and morphological phenomena (e.g., subject–verb agreement, anaphor binding, island constraints). Each paradigm consists of automatically generated sentence pairs, where one member is grammatically acceptable and the other violates a targeted linguistic constraint. BLiMP thus offers controlled contrasts to isolate specific syntactic distinctions. Second, the **COMPS** dataset [18] extends the minimal-pair methodology to conceptual and semantic phenomena, focusing on compositional reasoning beyond surface syntax. Together, BLiMP and COMPS provide a broad coverage of linguistic contrasts, from morpho-syntactic agreement to semantic composition, enabling a systematic investigation of neuron-level selectivity across linguistic domains.

# B    Related Work

**Targeted syntactic evaluation and minimal pairs.**    Evaluating the grammatical competence of language models has evolved from calculating overall perplexity to using targeted diagnostic datasets. Early work introduced small-scale, hand-crafted test suites to verify specific syntactic generalizations [12, 14]. This methodology was significantly scaled up with benchmarks like BLiMP [24] and automated platforms such as SyntaxGym [4], which use minimal pairs to isolate grammatical phenomena and reveal systematic gaps in LM performance. However, while these behavioral metrics effectively diagnose *what* linguistic rules a model violates, they treat the model as a black box, offering limited insight into *where* and *how* these distinctions are represented internally [6, 7].

**Representational structure and probing.**    To understand internal representations, the community turned to diagnostic classifiers, or "probes." seminal layer-wise analyses demonstrated that classical NLP pipeline steps, such as part-of-speech tagging and parsing, are naturally rediscovered in the hierarchical geometry of transformer representations [22, 11, 13, 8, 9]. Despite these insights, the probing paradigm faces significant methodological criticism. A core debate concerns whether a probe reveals the model's intrinsic knowledge or merely exploits the probe's own capacity to learn the task from the embeddings [10]. Theoretical work using information-theoretic criteria further suggests that probing results can be confounded by the ease of extracting information rather than its explicit presence [20]. These limitations motivate the need for *probe-free* diagnostics, like our proposed framework, which directly measure selectivity without the interference of auxiliary training.

**Mechanistic interpretability and neuron-level analyses.**    Moving beyond layer-wise trends, mechanistic interpretability aims to reverse-engineer model components into human-understandable algorithms. Recent studies have identified specific attention heads and MLP layers that function as key-value memories [5] or localized circuits for factual recall [16, 3]. At the finest granularity, individual neuron analyses have attempted to classify units in translation and morphology tasks [2, 1]. Our work bridges the gap between these neuron-level diagnostics and linguistic theory. Unlike prior work that often focuses on editing factual knowledge or broad concepts, we apply a rigorous linguistic lens (via minimal pairs) to organize neurons. We complement existing mechanistic accounts by (i) utilizing *paired*, controlled contrasts to eliminate confounding variables, and (ii) discovering functional groups of neurons that align with theoretical distinctions between syntax, semantics, and their interface.

