# OpenReview forum: "Neuron-Level Linguistic Selectivity in LLMs via a Classifier-Free Framework"
_NeurIPS.cc/2025/Workshop/UniReps — UniReps2025 oral_

### Official Review · Reviewer_J7KC · 2025-09-03
**Nice paper with interesting findings about neurons specialisation -- recommend acceptance**

**Confidence:** 4

**Review:**

In this paper, the authors use minimal pair datapoints to quantify to which extent neutrons in LLMs encode the difference between the acceptable vs non-acceptable member of the pair. In particular, they define a *separability index* based on neural activations in response to each member of the pair, where the higher the index the clearer the encoding separation of the pair members. Moreover, the authors obtain a layer-wise selectivity curve as well. The results show that: i) mid layers specialise on structural well-formedness while deeper layers shift towards conceptual aspects; ii) neurons sensitive to semantics-syntax interfaces do not saturate entire layers but occupy localized subsets, suggesting that this interface requires some degree of modularity and coordination between subspaces; iii) neurons can specialised to be domain-specific and domain-general.

**Workshop acceptance**: I recommend this paper for acceptance in the workshop. The reason why I give 2 instead of 3 is that I think the paper would benefit from a more systematic comparison with human processing strategies. It is not so necessary per se, however, if there, it would make this paper an ever better fit for the workshop.

**General comments**: I just have a minor question that I wanted to raise. I see that the authors collect activations for the last tokens of the minimal pairs. Is this choice motivated? Alternative, related approaches focus on getting the maximum activation across tokens (e.g., https://arxiv.org/abs/2110.02802). Maybe it would be good to comment on this aspect.

**Score:**

4

**Topic Fit:**

2

---

### Official Review · Reviewer_4rvT · 2025-09-06
**Review of this Paper**

**Confidence:** 4

**Review:**

The paper proposes a probe-free framework for neuron-level linguistic analysis in LLMs. Using minimal pairs (BLiMP, COMPS), the authors define a correlation-based Minimal-Pair Neuron Separability (MPNS) score to quantify how well neurons distinguish grammatical vs. ungrammatical inputs. Layer-wise selectivity curves and clustering reveal syntax-sensitive vs. semantics-sensitive neurons in Qwen3-0.6B.

**Strengths**:

Simple, reproducible, and probe-free; avoids confounds from auxiliary classifiers.

Combines minimal-pair evaluation with neuron-level analysis, offering an intuitive diagnostic tool.

**Weaknesses**:

Limited novelty: more a combination of existing ideas than a fundamentally new method.

Small experimental scope (only Qwen3-0.6B, no larger models or comparisons).

Lacks causal validation (e.g., neuron ablations).

**Score:**

3

**Topic Fit:**

3

---

### Official Review · Reviewer_XcTU · 2025-09-14
**Novel Framework for Neuron-Level Interpretability: Strong Contribution, Limited Scope**

**Confidence:** 3

**Review:**

Summary: The paper introduces Minimal-Pair Neuron Separability (MPNS), a probe-free correlation-based metric for assessing neuron-level linguistic selectivity in LLMs. By leveraging controlled minimal pairs from BLiMP (syntactic/morphological contrasts) and COMPS (conceptual/semantic contrasts), the authors compute separability scores that reveal where and how neurons differentiate grammatical vs. ungrammatical inputs. Applied to Qwen3-0.6B, the framework yields layer-wise selectivity curves and interpretable neuron clusters aligned with linguistic domains.

Pros:

	•	Novel contribution: The probe-free approach avoids classifier confounds and directly measures contrastive sensitivity, extending the toolkit for mechanistic interpretability.

	•	Methodological clarity: The MPNS score is mathematically simple, correlation-based, and scale-invariant (thanks to z-scoring).

	•	Findings: Results confirm known representational trends (early syntax/morphology vs. later semantics/concepts) while providing fine-grained neuron-level maps.

	•	Visualization: Figures (layer-wise curves and clustering heatmaps) effectively demonstrate both progression across layers and functional subpopulations.

	•	Interpretability: Clustering highlights both domain-specific neurons (syntax or concept specialists) and domain-general neurons integrating multiple domains—suggesting emergent modularity.


Cons:

	•	Single model scope: The analysis is restricted to Qwen3-0.6B; validation on larger or diverse models would strengthen generalizability.

	•	Hyperparameters: The fixed analysis layer (ℓ=15) and top-K choice (K=5) are reasonable but could benefit from more explicit justification or sensitivity analysis.

	•	Interpretation depth: While clustering results are promising, the discussion could elaborate further on implications for interventions or links to cognitive neuroscience.


Technical Strengths:

	•	Z-scoring ensures fair, paradigm-independent comparison.

	•	Correlation-based separability index is intuitive yet rigorous.

	•	Hierarchical clustering (average linkage) reveals coherent, interpretable neuron groups.

	•	Neuroscience inspiration (minimal pairs, functional localization, clustering) enriches the methodological grounding.


Minor Suggestions:

	•	Explicitly connect the layer progression finding (syntax early, semantics/concepts later) to prior literature on representational stages in LMs.

	•	Provide small-scale statistical validation (e.g., confidence intervals or random baselines) to support the robustness of peaks/clusters.

	•	Briefly outline future work: scaling to larger models, testing cross-lingual minimal pairs, or intervention experiments (e.g., neuron editing).


Verdict:

This is a strong methodological contribution that advances neuron-level interpretability of LLMs. The probe-free minimal-pair approach is both elegant and practical, yielding interpretable maps of linguistic selectivity. While limited to one model, the work convincingly demonstrates its utility and is well-suited for a NeurIPS extended abstract or workshop setting.

**Score:**

4

**Topic Fit:**

3